# Fabrication of Ordered Macropore Arrays in n-Type Silicon Wafer by Anodic Etching Using Double-Tank Electrochemical Cell

**DOI:** 10.3390/mi15050569

**Published:** 2024-04-26

**Authors:** Jing Zhang, Faqiang Zhang, Mingsheng Ma, Zhifu Liu

**Affiliations:** 1CAS Key Laboratory of Inorganic Functional Materials and Devices, Shanghai Institute of Ceramics, Chinese Academy of Sciences, Shanghai 201899, China; zhangjing20b@mails.ucas.ac.cn (J.Z.); zhangfq@mail.sic.ac.cn (F.Z.); mamingsheng@mail.sic.ac.cn (M.M.); 2College of Materials Science and Opto-Electronic Technology, University of Chinese Academy of Sciences, Beijing 100049, China

**Keywords:** macropore array, anodic etching, wafer thickness, etching voltage

## Abstract

In this work, ordered macropore arrays in n-type silicon wafers were fabricated by anodic etching using a double-tank electrochemical cell. The effects of the wafer thickness, etching time and voltage on the quality of macropore arrays were investigated. Homogeneous macropore arrays could be achieved in 200 μm thick silicon wafers, but could not be obtained from 300 and 400 μm thick silicon wafers. Highly ordered macropore arrays with an aspect ratio of 19 were fabricated in 200 μm thick n-type silicon at 4.5 V. The etching current decreases in 200 μm thick silicon but increases in thicker silicon with an increase in time. It demonstrates that the minority carrier transportation capability from the illuminated surface to the reactive surface is different for silicon wafers with different thicknesses. The minority carrier concentration at the illuminated surface for stable macropore formation and the current under different etching voltages were calculated based on a hole transport model. The results show that appropriately decreasing wafer thickness and increasing voltage can help stable macropore array fabrication in the illumination-limited double-tank cell.

## 1. Introduction

Since the macropore fabricated on n-type silicon was first reported in 1970’s [1], it has attracted great attention for many applications, such as photonic crystals [2,3,4,5,6], field effect transistors (FETs) [7,8], high-density silicon capacitors [9,10,11,12,13,14], templates [15], through silicon vias [16,17], microneedles [18,19,20,21], microfluidics [22,23], heat sinks [24], neutron detectors [25,26], and so on. Among all of them, the crucial problem is to fabricate controllable and homogeneous macropore arrays, especially for high-aspect-ratio ordered macropores. The controlled fabrication of macroporous silicon can lead to a regular pattern of uniform pores with minimal changes in the diameter both between neighboring pores and with depth [4,5,6]. There are two main ways to fabricate macropores: dry etching and wet etching. Dry etching usually removes materials from the substrate using reactive gases or plasmas and is commonly used in the semiconductor industry due to its higher selectivity and higher control on feature shapes, but the facility is very expensive for deep- and high-aspect-ratio etching and the sidewall scalloping pattern cannot be avoided in dry-etched macropores [27]. Meanwhile, wet etching is a chemical etching to dissolve materials by use of liquid chemicals or etchants without expensive equipment, and can be cost-effective to fabricate macroporous silicon materials, although it may have lower selectivity and limited control on feature shapes [28]. Electrochemical etching is the most popular method of wet etching. 

According to the amounts of cells, electrochemical etching can be classified into single-tank etching and double-tank etching. The single-tank etching is used widely, but requires good ohmic contacts on silicon by metal contact [29,30,31] or doping [32,33,34,35], which would cause wafer contamination. Moreover, macropore formation in the n-type silicon needs illumination to offer enough holes to induce etching reactions, which may cause electrolyte overheating in the single tank. However, using a double-tank electrochemical cell can replace the ohmic contacts with electrolyte contacts to avoid unnecessary contamination and control the electrolyte temperature conveniently. The double-tank etching is mostly used to fabricate irregular porous silicon [36,37,38] and p-type macroporous silicon [39,40]. Few investigations have been focused on the fabrication of n-type macroporous silicon, especially for the ordered macropore arrays. 

During the double-tank electrochemical etching, illumination needs to penetrate the liquid to irradiate the backside of silicon, which may cause the insufficiency of minority carriers for etching reaction and then increase inhomogeneity and roughness of macropore arrays in n-type silicon. To fabricate high-quality macropores, Meerakker [41] considered hole infusion by phosphorus diffusion, but the process may also cause wafer contamination, in the same way as single-tank electrochemical etching. Zhao [30] applied several halogen lamps to achieve strong illumination, which can increase the concentration of photo-generated holes to improve the etching efficiency and pore quality. But, too-strong illumination would heat up the cells and a cooling system is needed to keep stable etching. Another way is to use lightly-doped silicon with resistivity larger than kΩ·cm [25,26], but the diameters and wall thicknesses of macropores are usually larger than 10 μm which may define the limit of the aspect ratio. These works can improve hole concentrations in the n-type silicon wafer and enhance etching efficiency. But, little research focuses on optimizing hole transportation during the etching process to improve the macropore quality and aspect ratio. 

In this work, highly ordered macropore arrays were obtained in n-type silicon using a double-tank electrochemical etching method. The effects of wafer thickness, etching time, and voltage were studied. High-aspect-ratio macropore arrays were obtained. And, a hole transport model was developed based on the experimental results to understand the etching phenomena.

## 2. Materials and Methods

A double-tank electrochemical cell (Figure 1a) was used to fabricate n-type ordered macropore arrays. The cell body was two polytetrafluoroethylene (PTFE) chambers. Starting materials are n-type silicon wafers with different thicknesses of 200, 300, and 400 μm (resistivity 1–10 Ω·cm). And, a 200 nm thick silicon dioxide layer was deposited by low-pressure chemical vapor deposition on the surface of the silicon wafers, provided by Suzhou RDMICRO. Figure 1b shows the fabrication process used in this study. Firstly, the two-dimensional tetragonal lattice of squares with side 3 μm and pitch 8 μm was patterned on the silicon wafer surface by ultra-violet (UV) photolithography with a patterned mask. The positive photoresist (AZ-1500) was applied onto the wafer by spin coating (600 rpm, 6 s and 7000 rpm 60 s) followed by a softbake at 105 °C for 110 s. The samples were exposed for 22 s using a mask aligner (365 nm mask aligner, KE-MICRO, China) and then immersed in the ZX-238 developer for 16 s. After that, the samples were post-baked at 120 °C for 120 s. Then, the pattern was transferred to the silicon dioxide layer by inductively coupled plasmas (ICP) etching system (DISC-ICP-601, Beijing Chuangshiweina Technology Co., Ltd., Beijing, China) using SF_6_ plasma for 400 s. The upper and lower radio frequency (RF) power supplies were set to 200 W and 50 W, respectively, for the ICP etching. The values of gas flow and chamber pressure were kept constant at 60 sccm and 1 Pa. After photoresist removal, the pattern was transferred to the silicon surface by KOH (30 wt% water solution) etching through the patterned silicon dioxide. Thus, the patterned silicon wafer with inverse pyramid notches as the initiation sites was obtained. All silicon wafers were cleaned in piranha solution, HF solution, deionized water, acetone and ethanol in sequence, and then dried by N_2_. The cleaned silicon wafer would be mounted into the double-tank cell for electrochemical etching.

The etching electrolyte was a mixture of hydrofluoric acid (>40%HF in water), ethanol and water with a volume ratio of 3:7:20. The conductive electrolyte in another cell was saturated NaCl solution, which was pumped through a thermostat to control the temperature at 17 °C. The backside of the silicon wafer was illuminated by a 1000 W tungsten-halogen lamp. Potentiostatic etching method was used and the constant voltage between the Pt anode and cathode was controlled by CHI660c electrochemical workstation. The anode was immersed in the conductive electrolyte while the cathode was immersed in the etching electrolyte. Additionally, the size of Pt electrodes was a little larger than the wafer size. The etching time is about 120 min without a specific statement. After electrochemical etching, the samples were rinsed by deionized water and ethanol, and then dried by N_2_. Finally, the etched silicon wafers were cut and observed using Phenom Pro scanning electron microscope (SEM, Phenom World, Eindhoven, The Netherlands).

## 3. Results and Discussion

### 3.1. Effect of Wafer Thickness

We investigated the influence of silicon wafer thickness on the macropore formation. Figure 2a–c show the cross-sectional SEM images of silicon wafers with different thicknesses etched at 3.5 V for 120 min. Figure 2d shows the macropore depth and diameter with different wafer thicknesses. For the 400 μm thick silicon, the shape of the pores is irregular and dendritic, and the mean depth is just around 22.90 μm. The depth of macropores in the 300 μm thick silicon could be up to around 68.41 μm but is not uniform (standard deviation sd = 24.08 μm). Uniform and straight macropores could be obtained in the 200 μm thick silicon wafers with the mean depth of 78.38 μm (standard deviation sd = 0.72 μm), the mean diameter of 4.76 μm and the aspect ratio of 16.5 under the given condition. Figure 2e shows the etching current density-voltage (j-V) curves of silicon wafers with thicknesses of 200, 300 and 400 μm under backside illumination. The difference of the starting dissolution voltage may be due to the discrepancy of wafer and solution resistances [42] in each experiment. It is obvious that the starting dissolution voltages are about 2~2.5 V in our study. The dissolution current starts and then increases with increasing voltage when the applied voltage is higher than the starting dissolution voltage. Additionally, the slope of the etching current decreases with increasing wafer thickness because of the increased ohmic drop.

It is proposed that stable etching occurs at the local sites with a current density J_ps_, which is observable by a peak and a change in slope in the curve. When etching current density J < J_ps_, the reaction is limited by charge supply from the electrode, which means holes are depleted and HF accumulates, and then macropores are obtained. When J > J_ps_, the reaction is limited by mass transport in the electrolyte near the reactive surface, which means HF is depleted, holes accumulate, and then electropolishing occurs [29]. However, J_ps_ is not observed in the 300 μm and 400 μm thick silicon, and there is a current plateau instead. With wafer thickness decreasing, the saturated current under the high voltage would increase. When the thickness reduces to 200 μm, the peak of J_ps_ emerges. This discrepancy may be because of the limited illumination intensity that silicon could obtain in the double-tank electrochemical cell. Under this condition, the dissolution reaction on the silicon surface is determined by the supply of photo-generated holes rather than the charge transfer or chemical mass diffusion. The limitation of photo-generated hole supply would cause nonlinear hole diffusion through the whole wafer, resulting in a non-uniform reaction at the pore tips. From the experiment result, reducing the wafer thickness can mitigate this phenomenon.

Figure 2f shows etching current-time (I-t) curves during the etching process of wafers with different thicknesses. It can be seen that the current increases with decreasing wafer thickness. The recorded current decreases slightly with etching time for the 200 μm thick silicon wafers as the result of mass diffusion limitation with pore growth. While, for 300 and 400 μm thick silicon, the current increases with time. This may be because of the fact that, pores become closer to the backside of silicon with the etching progress and it is easier to collect more minority carriers. However, due to illumination limitation on silicon wafers in the double-tank electrochemical cell at the given condition, the reactive surface could not obtain enough holes to carry out a stable reaction so that irregular macropores are obtained in 300 and 400 μm thick silicon wafers.

### 3.2. Effect of Etching Time

Figure 3a,b show cross-sectional SEM images of the 200 μm and 300 μm thick silicon wafers etched at different times, respectively. The pore diameters and pore depths are shown in Figure 3c. The pore diameter and depth of two types of silicon wafers increase with time. But there is a difference in the increased amplitude: 300 μm thick silicon shows a larger increase in the amplitude of the pore diameter, and 200 μm thick silicon shows a larger increase in the amplitude of the pore depth. Additionally, within thirty minutes, homogeneous macropores occur in both types of silicon. However, the pore diameter and depth become inhomogeneous in the 300 μm thick silicon and the degree of inhomogeneity increases with time going by. Meanwhile, stable and homogeneous macropore arrays can be obtained in the 200 μm thick silicon for two hours. 

Let us look into the macropore formation process in silicon wafers of different thicknesses. Due to different wafer thicknesses, the concentration of minority carriers collected by pore tips is different. It is more likely to acquire a non-uniform concentration of holes at the pore tips for thick silicon wafers under limited illumination. Figure 3c shows pore diameter and pore depth synchronously increase with time. In the early stages, the pore is not deep so that pore growth could consume little minority carriers. As the pores grow, it needs more minority carriers to form macropore arrays, so the problem of insufficient photo-generated holes is obvious and the macropores begin to be inhomogeneous. The thicker the silicon is, the sooner the problem emerges. For 300 μm thick silicon, there is a non-uniform increase in pore diameter and depth within thirty minutes. However, it may occur only after etching three hours for 200 μm thick silicon under the same experimental condition. 

The j-V curves after different etching times are shown in Figure 3d,e. From the curves, there is a little decrease in the current density with etching time increasing for the 200 μm thick silicon wafers, but an obvious increase on the 300 μm thick silicon wafers, which are consistent with the results shown in Figure 2f. The current decrease in the 200 μm thick silicon wafer may be caused by the limitation of mass transport in the deep macropores. But, for 300 μm thick silicon wafers, the current is not saturated at 10 V even after two hours and the peak of J_ps_ could emerge. This means there is the possibility to fabricate stable macropore arrays if the supply of photo-generated holes increases to a certain value. We did not try higher illumination because of the facility limit. However, since the mean etching depth of 300 μm thick silicon wafer is about 70 μm, we can deduce that the silicon wafer with a thickness below 230 μm should be suitable for obtaining uniform macropore arrays. 

### 3.3. Effect of Etching Voltage

To optimize the etching process, we further studied the effect on the etching voltage at the etching time of 120 min. The SEM images are shown in Figure 4a–f, and the changes in pore diameter and depth are shown in Figure 4g. There is an evident increase in pore diameter and depth with etching voltage increasing, but the increase in pore depth becomes slower above 4.5 V, which means it may reach the maximum value. Figure 4h shows the etching current curves at different voltages, and the current increases as voltage increases. This means the high voltage can promote the collection of more minority carriers to boost the etching rate. However, when the voltage is too high, there is unexpected side-wall dissolution at the end of pores as shown in Figure 4f. Thus, choosing a suitable voltage is also important to obtain stable and deep macropore arrays. Based on these, the macropores with a maximum depth of 117.19 μm and an etching rate of 58.6 μm/h could be obtained at 5.5 V, and the macropores with a maximum aspect ratio of 19 could be obtained at 4.5 V in the 200 μm thick silicon wafer.

### 3.4. Stability Analysis on Minority Carrier Transport Model

From the previous experimental results, there is stable macropore formation or unstable macropore formation due to different conditions of minority carriers. When the concentration of minority carriers is sufficient, minority carriers are able to linearly diffuse to the pore tips and homogeneous macropore arrays can be obtained. However, when the concentration is insufficient, minority carriers would nonlinearly diffuse to the pore tips and inhomogeneous macropores would be formed, as shown in Figure 5a. The concentration of minority carriers is crucial. So, in this section, we calculated the concentration of holes needed for stable macropore formation and analyzed the effect of wafer thickness and etching voltage on the macropore arrays formation. 

We analyzed the minority carrier transportation according to the equation at the one-dimensional condition.
(1)$$D_{p}\frac{d^{2}\delta p}{dx^{2}}-\mu_{p}E\frac{d\delta p}{dx}+G_{p}\left(x\right)-\frac{\delta p}{\tau_{p}}=0$$
where $D_{p}$, $\mu_{p}$, δp and $\tau_{p}$ are the diffusion coefficient, mobility coefficient, the concentration and lifetime of the holes, respectively. And the generation rate $G_{p}\left(x\right)$ of electron-hole pairs at a distance x from the backside illuminated surface is given by [43]
(2)$$G_{p}\left(x\right)=P\alpha exp\left(-\alpha x\right)$$
where P is the total number of photons absorbed by the semiconductor per unit area and α is the absorption coefficient. 

The general solution of Equation (1) is
(3)$$\delta p=A_{1}e^{b_{+}x}{+A}_{2}e^{b_{-}x}-\frac{P\alpha \tau_{p}}{\alpha^{2}{L_{P}}^{2}+\mu_{p}\tau_{p}\alpha E-1}e^{-\alpha x}$$

The coefficients $b_{+}$ and $b_{-}$ are the roots of the characteristic quadratic equation
b±=E2kTe1±1+4kTeμpτpE2
with Dp=μpkTe. Additionally, k is the Boltzmann constant, T is the temperature and e is the elementary charge. The coefficients $A_{1}\;$ and $A_{2}\;$ are defined by the boundary conditions. The first one is $\frac{d\delta p}{dx}{|}_{x=0}=0$, because it can be regarded as no recombination at the illuminated surface. The other one is $\delta p{|}_{x=D-W}=0$, because minority carriers are assumed to be exhausted at the depletion edge on the condition of stable macropore array formation. And, at E>0, the electrical field coincides with the direction of minority diffusion, so that $b_{+}>0$, $b_{-}<0$. Based on these, the coefficients $A_{1}\;$ and $A_{2}$ are obtained
$$A_{1}=\frac{-P\alpha \tau_{p}\left\{\alpha \exp\left[b_{-}\left(D-W\right)\right]+b_{-}exp\left[-\alpha \left(D-W\right)\right]\right\}}{\left(\alpha^{2}L_{p}^{2}+\mu_{p}\tau_{p}\alpha E-1\right)\times \left\{b_{+}exp\left[k_{-}\left(D-W\right)\right]-b_{-}exp\left[b_{+}\left(D-W\right)\right]\right\}}$$
$$A_{2}=\frac{P\alpha \tau_{p}\left\{\alpha \exp\left[b_{+}\left(D-W\right)\right]+b_{+}exp\left[-\alpha \left(D-W\right)\right]\right\}}{\left(\alpha^{2}L_{p}^{2}+\mu_{p}\tau_{p}\alpha E-1\right)\times \left\{b_{+}exp\left[b_{-}\left(D-W\right)\right]-b_{-}exp\left[b_{+}\left(D-W\right)\right]\right\}}$$

The photocurrent through the silicon wafer is determined by the drift and diffusion components of the transfer of holes according to the following equation:(4)$$J=pq\mu_{p}E-qD_{p}\frac{dp}{dx}$$
where p=δp and $\frac{dp}{dx}=\frac{d\delta p}{dx}$, the dark concentration of holes is neglected. Taking Equation (3) into account, the photocurrent can be expressed by
(5)$$J=\left(\mu_{p}E-D_{p}b_{+}\right)qA_{1}e^{b_{+}x}+\left(\mu_{p}E-D_{p}b_{-}\right)qA_{2}e^{b_{-}x}-\left(\mu_{p}E+D_{p}\alpha \right)\frac{qP\alpha \tau_{p}}{\alpha^{2}{L_{P}}^{2}+\mu_{p}\tau_{p}\alpha E-1}e^{-\alpha x}$$

Based on Equation (5), we can calculate the stationary concentration of holes (*p*_0_) at the backside and the etching current of the silicon wafer. For ease of calculation, assuming that the silicon wafer is illuminated by monochromatic light with wavelength of 900 nm, the absorption coefficient α is 100 cm^−1^ and the total number of photons *P* is 10^19^ cm^−2^·s^−1^. Additionally, the diffusion coefficient and lifetime of minority carriers are 12 cm^2^/s and 10 μs, respectively [44]. In addition to this, the temperature is set as 290 K according to our experimental condition. The electrical field E is defined by the value which is the voltage divided by distance between the anode and cathode (15 cm in this work). 

Firstly, the stationary hole concentration at the backside of the silicon wafer with different thicknesses and pore depths was calculated, as shown in Figure 5b. At the beginning of etching, the highest hole concentration was needed for obtaining stable macropore formation. The stationary hole concentration decreases with decreasing wafer thickness. This means that, on the thinner silicon wafer, it is easier to find stationary macropore growth conditions, especially under the insufficient carrier supply condition. And with the pore growth, the distance between the illuminated surface and the reactive surface becomes closer. So, the ability to collect minority carriers at the end of the pore is enhanced, which may decrease the need for stationary hole concentration at the illuminated surface. But, with pore growth, mass transportation would be more difficult in the deeper pores and hole diffusion would not be the most effective factor for stable macropore formation. The chemical reaction and electrolyte diffusion need to be taken into consideration.

Figure 5c shows a current-voltage curve of 200 μm thick silicon wafer. The current almost linearly increases with the voltage in both experimental and calculated data, which means that high-aspect-ratio macropore arrays may be obtained in a short time by applying appropriately high voltage. However, the slope of the experimental data is slightly higher than that of the calculated data. The experimental value of the current is smaller than that of the calculated value. The deviation of slope may be because the hole consumption of sidewall dissolution is not considered in our calculation, and the high calculation current value may be due to our assumption based on the ideal condition where electric field uniformity, carrier recombination due to defects, and the electrochemical reaction were not taken into consideration. 

According to the calculation results, it makes sense that the wafer thickness and voltage are crucial factors in fabricating homogeneous and deep macropores. Wafer thickness decreasing means hole transport distance decreasing also, so the transportation efficiency would be heightened and the requirement for illumination would not be strict. And the increase in voltage means the increase in electric fields, whose direction is the same as the direction of hole diffusion. The hole drift velocity can be increased and more holes would be collected by reactive sites. However, according to our experimental results, high voltage would cause sidewall dissolution. This is because too many holes would deplete HF in the pores, so sidewall dissolution at the end of pores takes place, macropore arrays would be stripped from the bulk silicon, and the surface would be electropolished. Therefore, choosing an appropriate wafer thickness and etching voltage can help homogeneous macropore fabrication by optimizing hole transportation, especially in the condition of limited illumination.

## 4. Conclusions

The change in pore structures with respect to different wafer thicknesses, etching times, and voltages was investigated. The hole concentration and etching current at the stationary macropore formation process were theoretically calculated based on the hole transport model. The main findings of the paper are as follows:(1)Wafer thickness has a large effect on the backside hole concentration for stable macropores etching in n-type silicon. Decreasing wafer thickness can reduce hole concentration for stable macropore array formation.(2)Increasing voltage can promote hole transportation to obtain a higher etching current and etching rate.(3)Homogeneous macropore arrays with an aspect ratio of 19 were fabricated in 200 μm thick n-type silicon at 4.5 V by double-tank electrochemical etching.

This work provides a strategy for optimizing the hole transport process for stable macropore array formation in n-type silicon by electrochemical etching under limited illumination.

## Figures and Tables

**Figure 1 micromachines-15-00569-f001:**
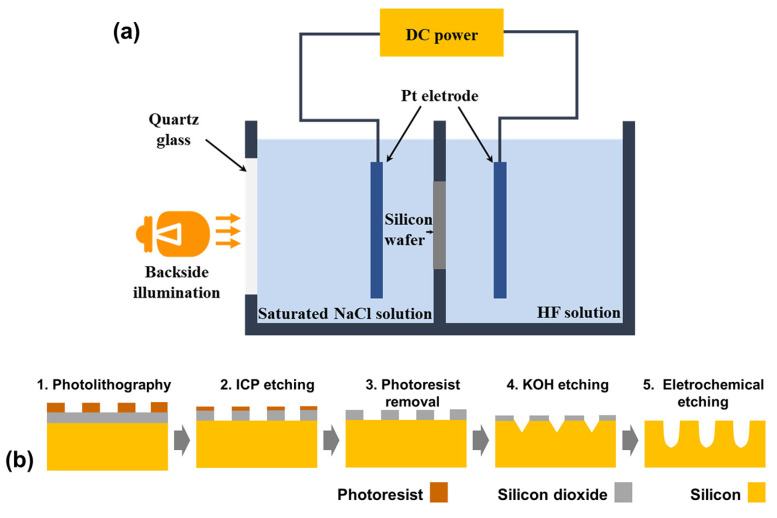
(**a**) Schematic diagram of double-tank electrochemical cell. (**b**) Schematic steps for ordered macropore fabrication process.

**Figure 2 micromachines-15-00569-f002:**
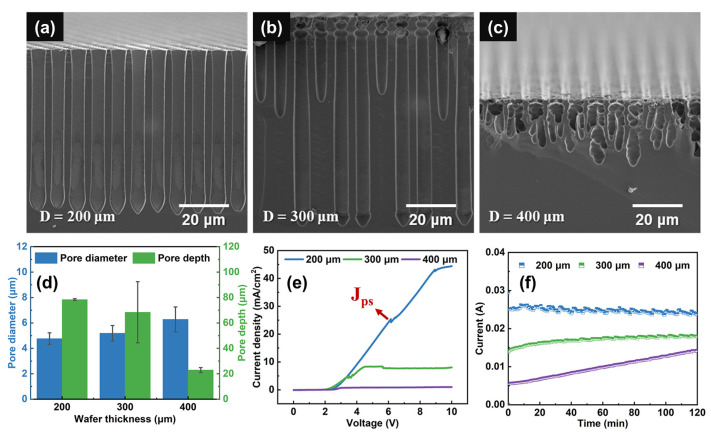
Cross-sectional SEM images of macropore arrays fabricated on the n-type silicon with the thickness of (**a**) 200 μm, (**b**) 300 μm, (**c**) 400 μm. (**d**) Pore diameter and depth vs. wafer thickness. (**e**) The j-V curves of n-type silicon wafers with different thicknesses. (**f**) I-t curves of macropore fabrication using wafers with different thicknesses. The etching voltage is 3.5 V.

**Figure 3 micromachines-15-00569-f003:**
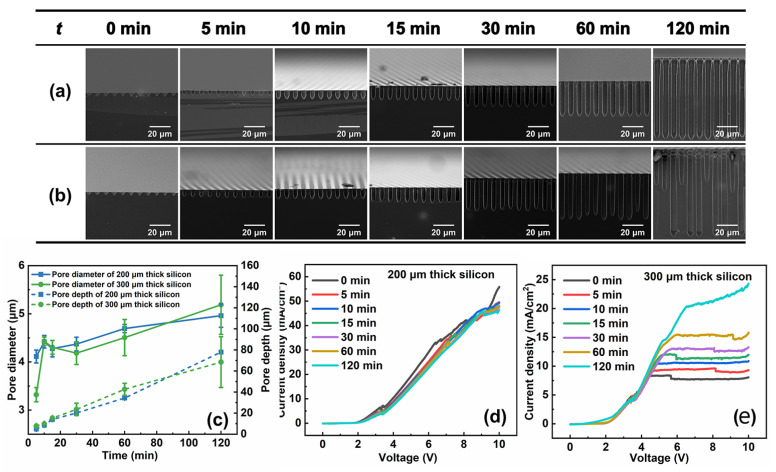
Cross-sectional SEM images of macropore arrays etched at different times on the n-type silicon wafer with thicknesses of (**a**) 200 μm and (**b**) 300 μm. (**c**) Experimental data on pore diameter and pore depth vs. time. The etching voltage is 3.5 V. j-V curves of 200 μm and 300 μm silicon wafers at different etching times were shown in (**d**) and (**e**), respectively.

**Figure 4 micromachines-15-00569-f004:**
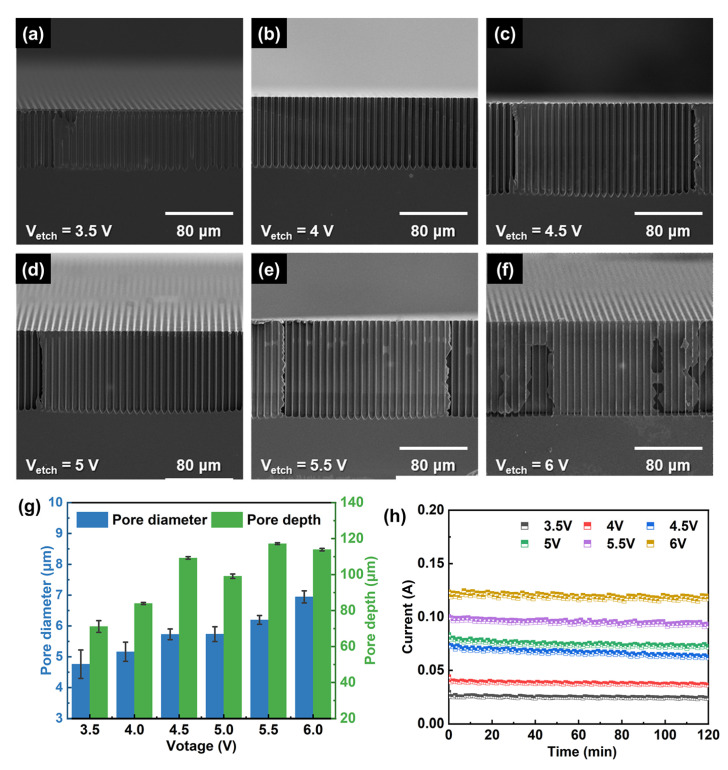
Cross-sectional SEM images of macropore arrays etched in 200 μm thick silicon wafers at different etching voltage of (**a**) 3.5 V, (**b**) 4 V, (**c**) 4.5 V (**d**) 5 V, (**e**) 5.5 V, and (**f**) 6 V. (**g**) Pore diameter and depth vs. etching voltage. (**h**) I-t curves of macropore fabricated at the different voltages.

**Figure 5 micromachines-15-00569-f005:**
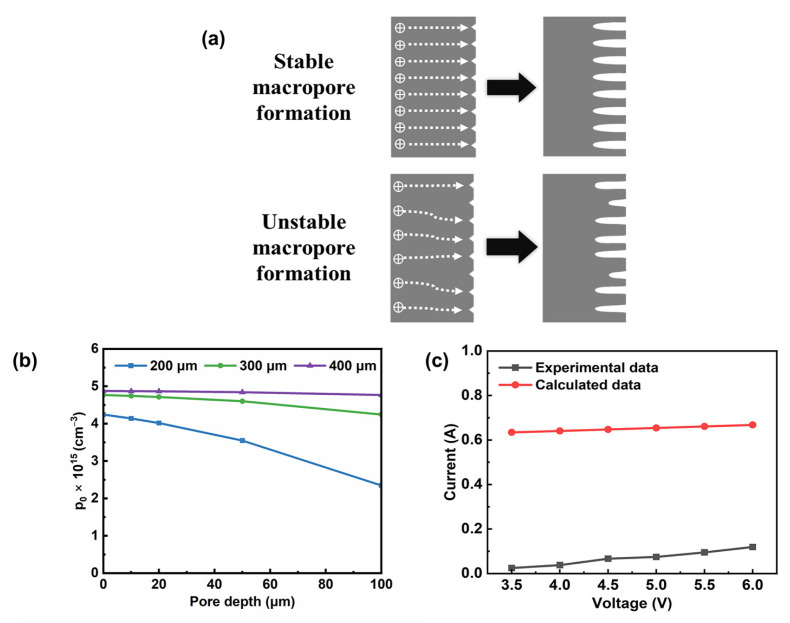
(**a**) Diagram of stable macropore formation and unstable macropore formation. (**b**) Calculated data of the stationary minority carrier concentration at the illuminated surface vs. pore depth (E=0.23 V/cm). (**c**) Calculated data and experimental data of the current under voltage of 3.5 V (E = 0.23 V/cm), 4.0 V (E = 0.27 V/cm), 4.5 V (E = 0.3 V/cm), 5.0 V (E = 0.33 V/cm), 5.5 V (E = 0.36 V/cm) and 6.0 V (E = 0.4 V/cm) on the 200 μm thick silicon wafers.

## Data Availability

All the relevant data are included in this published article.

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
