# Peer review of "Fabrication of Ordered Macropore Arrays in n-Type Silicon Wafer by Anodic Etching Using Double-Tank Electrochemical Cell"

_micromachines, 2024, doi:10.3390/mi15050569_

Round 1
Reviewer 1 Report
Comments and Suggestions for Authors
Nice paper, accepted after a minor revision.
The materials and Methods section must include:
1. The deposition method of the SiO2 on the silicon surface.
2. The duration range of the potentiostatic etching.
Results and Discussion section must include:
1. Change «Fig. 2d shows the macropore depth and diameter of silicon 104 wafers with different thicknesses» to «Fig. 2d shows the macropore depth and diameter with different wafers thicknesses».
2. Change “…the mean diameter of 4.76 and the aspect ratio of 16.5 under the given condition” to “…the mean diameter of 4.76 mm and the aspect ratio of 16.5 under the given condition”.
3. What is the value Jps? Indicate Jps in the figure 2e.
4. At 60 minutes of etching for a 300 µm wafer, the pore depth is 35 µm (fig.3b). At 120 minutes of etching, the pore depth should either remain the same or increase, but not decrease. In your case, after 120 minutes of etching, the thickness of some pores increases, and some decreases to 26 microns (fig.3b). Clarify, please.
5. Line 251. You must put a period at the end of the sentence.
Reference
1. Reference 39. Correct spelling of the reference: “Onyshchenko, V.F.; Karachevtseva, L.A. Effective Minority Carrier Lifetime and Distribution of Steady State Excess Minority Carriers in Macroporous Silicon. Him. Fiz. Tehnol. Poverhni 2017, 8, 322–332, doi:10.15407/hftp08.03.322”.
Author Response
Thank you very much for taking the time to review this manuscript. Please find the detailed responses in the attachment and the corresponding revisions have been marked in red in the re-submitted files.

Reviewer 2 Report
Comments and Suggestions for Authors 1. The abstract misses the point. First of all, the authors did not indicate the main uses of macroporous silicon. Secondly, they did not indicate the parameters of the study. Finally, they did not indicate the best etching effect.2. When the authors mentioned dry etching and wet etching, in order to facilitate the reader's understanding, it is necessary to give the definition of dry etching and wet etching. In addition, the authors pointed out that the equipment for dry etching was expensive, but the advantages of dry etching as a commonly used etching method needed to be explained. For wet etching, although the equipment was cheaper and could efficiently and economically prepare macroporous silicon materials, the authors also needed to point out what its disadvantages were.
3. The authors pointed out that double-tank etching was mainly used to prepare irregularly porous silicon and p-type macroporous silicon. However, there was little research on the preparation of n-type macroporous silicon by using double-tank etching, especially the ordered macroporous arrays. Why is that? Is it because of some technical difficulties? Or is it because the use of double-tank etching is not the optimal preparation method? Here the authors needs to elaborate.
4. Please note that the statement is consistent. It should be a “double-tank etching”, not a “double-tank cells” in the second paragraph of the introduction. I think it's better to use “double-tank etching”. Otherwise it's not easy for the reader to understand.
5. In the third paragraph of the introduction, the authors pointed out that many current studies focus on improving the etching efficiency, while they wanted to study improving the etching quality. This is a good idea, however, their research should not only focus on the etching quality, they should also supplement the study of etching efficiency. In addition, they should compare the etching efficiency and quality of their research with that of other scholars to reflect the innovation of the research results of this paper.
6. What are UV photolithography and ICP etching? It is recommended not to use abbreviations directly. In addition, please indicate the detailed equipment and specific parameters used for UV photolithography and ICP etching.
7. The font size of serial numbers a, b and “DC power” in Figure 1 is too large, and there is an redundant full stop at the end of note 1(a). In addition, similar problems exist in Figures 2-5. Change the font size of the serial number.
8. By analyzing the influences of wafer thickness, etching time and voltage on the etching quality, the author obtained the best etching quality (Homogeneous macropore arrays with aspect ratio of 19). What is the current level of such etching quality, please explain by comparing it with other scholars. Whether it can reach the international level.
9. It is suggested that the conclusion should be divided into sections, so as to be more organized.
10. The references are too old, and the references of the past two years are lacking.
Based on the above comments on the author's paper, I think this paper should be major revision.
Comments on the Quality of English LanguagePlease check the language carefully and pay attention to grammar and diction.
Author Response

(The authors gave the same response as above.)

Reviewer 3 Report
Comments and Suggestions for Authors
The research are interesting: the effects of wafer thickness, etch time and voltge are investigated.
The work is of practical importance provides a strategy of optimizing hole transport process for stable macro-pore array formation in n-type silicon by electrochemical etching under limited illumination.
The work finds its place in the pages of the journal.
Author Response
Thanks for your encouraging comments for this work.

Round 2
Reviewer 2 Report
Comments and Suggestions for Authors
Nice! It has been modified according to the reviewer's opinion.